# Structured Prediction with Projection Oracles

**Mathieu Blondel**
NTT Communication Science Laboratories
Kyoto, Japan
mathieu@mblondel.org

## Abstract

We propose in this paper a general framework for deriving loss functions for structured prediction. In our framework, the user chooses a convex set including the output space and provides an oracle for *projecting* onto that set. Given that oracle, our framework automatically generates a corresponding convex and smooth loss function. As we show, adding a projection as output layer provably makes the loss smaller. We identify the marginal polytope, the output space's convex hull, as the best convex set on which to project. However, because the projection onto the marginal polytope can sometimes be expensive to compute, we allow to use any convex superset instead, with potentially cheaper-to-compute projection. Since efficient projection algorithms are available for numerous convex sets, this allows us to construct loss functions for a variety of tasks. On the theoretical side, when combined with *calibrated decoding*, we prove that our loss functions can be used as a consistent surrogate for a (potentially non-convex) target loss function of interest. We demonstrate our losses on label ranking, ordinal regression and multilabel classification, confirming the improved accuracy enabled by projections.

## 1   Introduction

The goal of supervised learning is to learn a mapping that links an input to an output, using examples of such pairs. This task is noticeably more difficult when the output objects have a structure, i.e., when they are not mere vectors. This is the so-called structured prediction setting [4] and has numerous applications in natural language processing, computer vision and computational biology.

We focus in this paper on the surrogate loss framework, in which a convex loss is used as a proxy for a (potentially non-convex) target loss of interest. Existing convex losses for structured prediction come with different trade-offs. On one hand, the structured perceptron [16] and hinge [52] losses only require access to a maximum a-posteriori (MAP) oracle for finding the highest-scoring structure, while the conditional random field (CRF) [29] loss requires access to a marginal inference oracle, for evaluating the expectation under a Gibbs distribution. Since marginal inference is generally considered harder than MAP inference, for instance containing #P-complete counting problems, this makes the CRF loss less widely applicable. On the other hand, unlike the structured perceptron and hinge losses, the CRF loss is smooth, which is crucial for fast convergence, and comes with a probabilistic model, which is important for dealing with uncertainty. Unfortunately, when combined with MAP decoding, these losses are typically inconsistent, meaning that their optimal estimator does not converge to the target loss function's optimal estimator. Recently, several works [15, 26, 39, 31] showed good results and obtained consistency guarantees by combining a simple squared loss with **calibrated decoding**. Since these approaches only require a decoding oracle at test time and no oracle at train time, this questions whether structural information is even beneficial during training.

In this paper, we propose loss functions for structured prediction using a different kind of oracle: **projections**. Kullback-Leibler projections onto various polytopes have been used to derive online algorithms [24, 56, 49, 1] but it is not obvious how to extract a loss from these works. In our

framework, the user chooses a convex set containing the output space and provides an oracle for projecting onto that set. Given that oracle, we automatically generate an associated loss function. As we show, incorporating a projection as output layer provably makes the loss smaller. We identify the marginal polytope, the output space's convex hull, as the best convex set on which to project. However, because the projection onto the marginal polytope can sometimes be expensive to compute, we allow to use instead any convex superset, with potentially cheaper-to-compute projection. When using the marginal polytope as the convex set, our loss comes with an implicit probabilistic model. Our contributions are summarized as follows:

- Based upon Fenchel-Young losses [11, 12], we introduce projection-based losses in a broad setting. We give numerous examples of useful convex polytopes and their associated projections.

- We study the **consistency** w.r.t. a target loss of interest when combined with calibrated decoding, extending a recent analysis [38] to the more general projection-based losses. We exhibit a **trade-off** between computational cost and statistical estimation.

- We demonstrate our losses on label ranking, ordinal regression and multilabel classification, confirming the improved accuracy enabled by projections.

**Notation.** We denote the probability simplex by $\triangle^p := \{q \in \mathbb{R}_+^p : \|q\|_1 = 1\}$, the domain of $\Omega : \mathbb{R}^p \to \mathbb{R} \cup \{\infty\}$ by $\mathrm{dom}(\Omega) := \{u \in \mathbb{R}^p : \Omega(u) < \infty\}$, the Fenchel conjugate of $\Omega$ by $\Omega^*(\theta) := \sup_{u \in \mathrm{dom}(\Omega)} \langle u, \theta \rangle - \Omega(u)$. We denote $[k] := \{1, \dots, k\}$.

## 2 Background and related work

**Surrogate loss framework.** The goal of structured prediction is to learn a mapping $f : \mathcal{X} \to \mathcal{Y}$, from an input $x \in \mathcal{X}$ to an output $y \in \mathcal{Y}$, minimizing the expected target risk

$$\mathcal{L}(f) := \mathbb{E}_{(X,Y) \sim \rho} \, L(f(X), Y),$$

where $\rho \in \triangle(\mathcal{X} \times \mathcal{Y})$ is a typically unknown distribution and $L : \mathcal{Y} \times \mathcal{Y} \to \mathbb{R}_+$ is a potentially non-convex target loss. We focus in this paper on surrogate methods, which attack the problem in two main phases. During the training phase, the labels $y \in \mathcal{Y}$ are first mapped to $\varphi(y) \in \Theta$ using an **encoding** or embedding function $\varphi : \mathcal{Y} \to \Theta$. In this paper, we focus on $\Theta := \mathbb{R}^p$, but some works consider general Hilbert spaces [15, 26, 31]. In most cases, $\varphi(y)$ will be a **zero-one encoding** of the parts of $y$, i.e., $\varphi(y) \in \{0, 1\}^p$. Given a surrogate loss $S : \Theta \times \Theta \to \mathbb{R}_+$, a model $g : \mathcal{X} \to \Theta$ (e.g., a neural network or a linear model) is then learned so as to minimize the surrogate risk

$$\mathcal{S}(g) := \mathbb{E}_{(X,Y) \sim \rho} \, S(g(X), \varphi(Y)).$$

This allows to leverage the usual empirical risk minimization framework in the space $\Theta$. During the prediction phase, given an input $x \in \mathcal{X}$, a model prediction $\theta = g(x) \in \Theta$ is "pulled back" to a valid output $\widehat{y} \in \mathcal{Y}$ using a **decoding** function $d : \Theta \to \mathcal{Y}$. This is summarized in the following diagram:

$$x \in \mathcal{X} \xrightarrow[\text{model}]{g} \theta \in \Theta \xrightarrow[\text{decoding}]{d} \widehat{y} \in \mathcal{Y}. \tag{1}$$

Commonly used decoders include the pre-image oracle [55, 17, 25] $\theta \mapsto \mathrm{argmin}_{y \in \mathcal{Y}} S(\theta, \varphi(y))$ and the maximum a-posteriori inference oracle [16, 52, 29], which finds the highest-scoring structure:

$$\mathrm{MAP}(\theta) := \underset{y \in \mathcal{Y}}{\mathrm{argmax}} \langle \theta, \varphi(y) \rangle. \tag{2}$$

In the remainder of this paper, for conciseness, we will use use $S(\theta, y)$ as a shorthand for $S(\theta, \varphi(y))$ but it is useful to bear in mind that surrogate losses are always really defined over vector spaces.

**Examples of surrogate losses.** We now review classical examples of loss functions that fall within that framework. The structured perceptron [16] loss is defined by

$$S_{\mathrm{SP}}(\theta, y) := \max_{y' \in \mathcal{Y}} \, \langle \theta, \varphi(y') \rangle - \langle \theta, \varphi(y) \rangle. \tag{3}$$

Clearly, it requires a MAP inference oracle at training time in order to compute subgradients w.r.t. $\theta$. The structured hinge loss used by structured support vector machines [52] is a simple variant of (3) using an additional loss term. Classically, it is assumed that this term satisfies an affine decomposition, so that we only need a MAP oracle. The conditional random fields (CRF) [29] loss, on the other

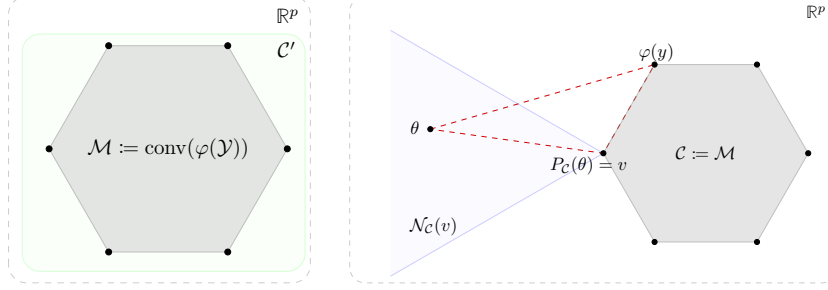

Figure 1: Proposed framework in the Euclidean geometry. **Left.** Each black point represents the vector encoding $\varphi(y)$ of one possible structure $y \in \mathcal{Y}$. We require to choose a convex set $\mathcal{C}$ including the encoded output space, $\varphi(\mathcal{Y})$. The best choice is $\mathcal{M}$, the convex hull of $\varphi(\mathcal{Y})$, but we can use any superset $\mathcal{C}'$ of it with potentially cheaper-to-compute projection. Setting $\mathcal{C} = \mathbb{R}^p$, our loss $S_{\mathcal{C}}(\theta, y)$ (omitting the superscript $\Psi$) recovers the squared loss (i.e., no projection). **Right.** When $\theta$ belongs to the interior of $\mathcal{N}_{\mathcal{C}}(v)$, the normal cone of $\mathcal{C}$ at a vertex $v$, the projection $P_{\mathcal{C}}(\theta) \coloneqq \operatorname{argmin}_{u \in \mathcal{C}} \|u - \theta\|_2$ hits the vertex $v$ and the angle formed by $\theta$, $P_{\mathcal{C}}(\theta)$ and $\varphi(y)$ is obtuse. In this case, $S_{\mathcal{C}}(\theta, y)$ is a strict upper-bound for $\ell_{\mathcal{C}}(\theta, y) \coloneqq \frac{1}{2} \|\varphi(y) - P_{\mathcal{C}}(\theta)\|_2^2$. When $\theta$ is not in the normal cone of $\mathcal{C}$ at any vertex, then the angle is right and the two losses coincide, $S_{\mathcal{C}}(\theta, y) = \ell_{\mathcal{C}}(\theta, y)$.

hand, requires a so-called marginal inference oracle [54], for evaluating the expectation under the Gibbs distribution $p(y; \theta) \propto e^{\langle \theta, \varphi(y) \rangle}$. The loss and the oracle are defined by

$$S_{\mathrm{crf}}(\theta, y) \coloneqq \log \sum_{y' \in \mathcal{Y}} e^{\langle \theta, \varphi(y') \rangle} - \langle \theta, \varphi(y) \rangle \quad \text{and} \quad \mathrm{marginal}(\theta) \coloneqq \mathbb{E}_{Y \sim p}[\varphi(Y)] \propto \sum_{y \in \mathcal{Y}} e^{\langle \theta, \varphi(y) \rangle} \varphi(y).$$

When $\varphi(y)$ is a zero-one encoding of the parts of $y$ (i.e., a bit vector), $\mathrm{marginal}(\theta)$ can be interpreted as some marginal distribution over parts of the structures. The CRF loss is smooth and comes with a probabilistic model, but its applicability is hampered by the fact that marginal inference is generally harder than MAP inference. This is for instance the case for permutation-prediction problems, where exact marginal inference is intractable [53, 50, 44] but MAP inference can be computed exactly.

**Consistency.** When working with surrogate losses, an important question is whether the surrogate and target risks are consistent, that is, whether an estimator $g^\star$ minimizing $\mathcal{S}(g)$ produces an estimator $d \circ g^\star$ minimizing $\mathcal{L}(f)$. Although this question has been widely studied in the multiclass setting [58, 6, 51, 35] and in other specific settings [21, 45], it is only recently that it was studied in a fully general structured prediction setting. The structured perceptron, hinge and CRF losses are generally not consistent when using MAP as decoder $d$ [38]. Inspired by kernel dependency estimation [55, 17, 25], several works [15, 26, 31] showed good empirical results and proved consistency by combining a squared loss $S_{\mathrm{sq}}(\theta, y) \coloneqq \frac{1}{2} \|\varphi(y) - \theta\|_2^2$ with calibrated decoding (no oracle is needed during training). A drawback of this loss, however, is that it does not make use of the output space $\mathcal{Y}$ during training, ignoring precious structural information. More recently, the consistency of the CRF loss in combination with calibrated decoding was analyzed in [38].

## 3 Structured prediction with projection oracles

In this section, we build upon Fenchel-Young losses [11, 12] to derive a class of smooth loss functions leveraging structural information through a different kind of oracle: **projections**. Our losses are applicable to a large variety of tasks (including permutation problems, for which CRF losses are intractable) and have consistency guarantees when combined with calibrated decoding (cf. §5).

**Fenchel-Young losses.** The aforementioned perceptron, hinge and CRF losses all belong to the class of Fenchel-Young losses [11, 12]. The Fenchel-Young loss generated by $\Omega$ is defined by

$$S_\Omega(\theta, y) \coloneqq \Omega^*(\theta) + \Omega(\varphi(y)) - \langle \theta, \varphi(y) \rangle. \tag{4}$$

As shown in [11, 12], $S_\Omega(\theta, y)$ satisfies the following desirable properties:

- Non-negativity: $S_\Omega(\theta, y) \geq 0$,
- Zero loss: $S_\Omega(\theta, y) = 0 \Leftrightarrow \nabla \Omega^*(\theta) = \varphi(y)$,

- Convexity: $S_\Omega(\theta, y)$ is convex in $\theta$,
- Smoothness: If $\Omega$ is $\frac{1}{\beta}$-strongly convex, then $S_\Omega(\theta, y)$ is $\beta$-smooth,
- Gradient as residual (generalizing the squared loss): $\nabla_\theta S_\Omega(\theta, y) = \nabla\Omega^*(\theta) - \varphi(y)$.

In the Fenchel duality perspective, $\theta = g(x)$ belongs to the dual space $\mathrm{dom}(\Omega^*) = \Theta = \mathbb{R}^p$ and is thus unconstrained. This is convenient, as this places no restriction on the model outputs $\theta = g(x)$. On the other hand, $\varphi(y)$ belongs to the primal space $\mathrm{dom}(\Omega)$, which must include the encoded output space $\varphi(\mathcal{Y})$, i.e., $\varphi(\mathcal{Y}) \subseteq \mathrm{dom}(\Omega)$, and is typically constrained. The gradient $\nabla\Omega^*$ is a mapping from $\mathrm{dom}(\Omega^*)$ to $\mathrm{dom}(\Omega)$ and $S_\Omega$ can be seen as loss with mixed arguments, between these two spaces. The theory of Fenchel-Young loss was recently extended to infinite spaces in [34].

**Projection-based losses.** Let the Bregman divergence generated by $\Psi$ be defined as $D_\Psi(u, v) \coloneqq \Psi(u) - \Psi(v) - \langle \nabla\Psi(v), u - v \rangle$. The Bregman projection of $\nabla\Psi^*(\theta)$ onto a closed convex set $\mathcal{C}$ is

$$P_\mathcal{C}^\Psi(\theta) \coloneqq \operatorname*{argmin}_{u \in \mathcal{C}} D_\Psi(u, \nabla\Psi^*(\theta)). \tag{5}$$

Intuitively, $\nabla\Psi^*$ maps the unconstrained predictions $\theta = g(x)$ to $\mathrm{dom}(\Psi)$, ensuring that the Bregman projection is well-defined. Let us define the Kullback-Leibler divergence by $\mathrm{KL}(u, v) \coloneqq \sum_i u_i \log \frac{u_i}{v_i} - \sum_i u_i + \sum_i v_i$. Two examples of generating function $\Psi$ are $\Psi(u) = \frac{1}{2}\|u\|_2^2$ with $\mathrm{dom}(\Psi) = \mathbb{R}^p$ and $\nabla\Psi^*(\theta) = \theta$, and $\Psi(u) = \langle u, \log u \rangle$ with $\mathrm{dom}(\Psi) = \mathbb{R}_+^p$ and $\nabla\Psi^*(\theta) = e^{\theta-1}$. This leads to the Euclidean projection $\mathrm{argmin}_{u \in \mathcal{C}} \|u - \theta\|_2$ and the KL projection $\mathrm{argmin}_{u \in \mathcal{C}} \mathrm{KL}(u, e^{\theta-1})$, respectively.

Our key insight is to use a projection onto a chosen convex set $\mathcal{C}$ as **output layer**. If $\mathcal{C}$ contains the encoded output space, i.e., $\varphi(\mathcal{Y}) \subseteq \mathcal{C}$, then $\varphi(y) \in \mathcal{C}$ for any ground truth $y \in \mathcal{Y}$. Therefore, if $\nabla\Psi^*(\theta) \notin \mathcal{C}$, then $P_\mathcal{C}^\Psi(\theta)$ is necessarily a better prediction than $\nabla\Psi^*(\theta)$, since it is closer to $\varphi(y)$ in the sense of $D_\Psi$. If $\nabla\Psi^*(\theta)$ already belongs to $\mathcal{C}$, then $P_\mathcal{C}^\Psi(\theta) = \nabla\Psi^*(\theta)$ and thus $P_\mathcal{C}^\Psi(\theta)$ is as good as $\nabla\Psi^*(\theta)$. To summarize, we have $D_\Psi(\varphi(y), P_\mathcal{C}^\Psi(\theta)) \leq D_\Psi(\varphi(y), \nabla\Psi^*(\theta))$ for all $\theta \in \Theta$ and $y \in \mathcal{Y}$. Therefore, it is natural to choose $\theta$ so as to minimize the following **compositional** loss

$$\ell_\mathcal{C}^\Psi(\theta, y) \coloneqq D_\Psi(\varphi(y), P_\mathcal{C}^\Psi(\theta)).$$

Unfortunately, $\ell_\mathcal{C}^\Psi$ is non-convex in $\theta$ in general, and $\nabla_\theta \ell_\mathcal{C}^\Psi(\theta, y)$ requires to compute the Jacobian of $P_\mathcal{C}^\Psi(\theta)$, which could be difficult, depending on $\mathcal{C}$. Other works have considered the output of an optimization program as input to a loss [48, 20, 8] but these methods are non-convex too and typically require unrolling the program's iterations. We address these issues, using Fenchel-Young losses.

**Convex upper-bound.** We now set the generating function $\Omega$ of the Fenchel-Young loss (4) to $\Omega = \Psi + I_\mathcal{C}$, where $I_\mathcal{C}$ denotes the indicator function of $\mathcal{C}$. We assume that $\Psi$ is Legendre type [46, 54], meaning that it is strictly convex and $\nabla\Psi$ explodes at the boundary of the interior of $\mathrm{dom}(\Psi)$. This assumption is satisfied by both $\Psi(u) = \frac{1}{2}\|u\|_2^2$ and $\Psi(u) = \langle u, \log u \rangle$. With that assumption, as shown in [11, 12], we obtain $\nabla\Omega^*(\theta) = P_\mathcal{C}^\Psi(\theta)$ for all $\theta \in \Theta$, allowing us to use Fenchel-Young losses. For brevity, let us define the Fenchel-Young loss generated by $\Omega = \Psi + I_\mathcal{C}$ as

$$S_\mathcal{C}^\Psi(\theta, y) \coloneqq S_{\Psi+I_\mathcal{C}}(\theta, y). \tag{6}$$

From the properties of Fenchel-Young losses, we have $S_\mathcal{C}^\Psi(\theta, y) = 0 \Leftrightarrow P_\mathcal{C}^\Psi(\theta) = \varphi(y)$ and $\nabla_\theta S_\mathcal{C}^\Psi(\theta, y) = P_\mathcal{C}^\Psi(\theta, y) - \varphi(y)$. Moreover, as shown in [11, 12], $S_\mathcal{C}^\Psi(\theta, y)$ upper-bounds $\ell_\mathcal{C}^\Psi(\theta, y)$:

$$\ell_\mathcal{C}^\Psi(\theta, y) \leq S_\mathcal{C}^\Psi(\theta, y) \quad \forall \theta \in \Theta, y \in \mathcal{Y}. \tag{7}$$

Note that if $\mathcal{C} = \mathrm{dom}(\Psi)$ (largest possible set), then $S_\mathcal{C}^\Psi(\theta, y) = D_\Psi(\varphi(y), \nabla\Psi^*(\theta))$. In particular, with $\Psi = \frac{1}{2}\|\cdot\|_2^2$ and $\mathcal{C} = \mathbb{R}^p$, $S_\mathcal{C}^\Psi(\theta, y)$ recovers the squared loss $S_\mathrm{sq}(\theta, y) = \frac{1}{2}\|\varphi(y) - \theta\|_2^2$.

**Choosing the projection set.** Recall that $\mathcal{C}$ should be a convex set such that $\varphi(\mathcal{Y}) \subseteq \mathcal{C}$. The next new proposition, a simple consequence of (4), gives an argument in favor of using smaller sets.

---

**Proposition 1** *Using smaller sets results in smaller loss*

*Let $\mathcal{C}, \mathcal{C}'$ be two closed convex sets such that $\mathcal{C} \subseteq \mathcal{C}' \subseteq \mathrm{dom}(\Psi)$. Then,*

$$S_\mathcal{C}^\Psi(\theta, y) \leq S_{\mathcal{C}'}^\Psi(\theta, y) \quad \forall \theta \in \Theta, y \in \mathcal{Y}.$$

---

As a corollary, combined with (7), we have
$$\ell_{\mathcal{C}}^{\Psi}(\theta, y) \leq S_{\mathcal{C}}^{\Psi}(\theta, y) \leq D_{\Psi}(\varphi(y), \nabla \Psi^*(\theta))$$
and in particular when $\Psi(u) = \frac{1}{2}\|u\|_2^2$, noticing that $S_{\mathrm{sq}} = S_{\mathbb{R}^p}^{\Psi}$, we have
$$\ell_{\mathcal{C}}^{\Psi}(\theta, y) = \frac{1}{2}\|\varphi(y) - P_{\mathcal{C}}^{\Psi}(\theta)\|_2^2 \leq S_{\mathcal{C}}^{\Psi}(\theta, y) \leq \frac{1}{2}\|\varphi(y) - \theta\|_2^2 = S_{\mathrm{sq}}(\theta, y).$$

Therefore, the Euclidean projection $P_{\mathcal{C}}^{\Psi}(\theta)$ always achieves a smaller squared loss than $\theta = g(x)$. This is intuitive, as $\mathcal{C}$ is a smaller region than $\mathbb{R}^p$ and $\mathcal{C}$ is guaranteed to include the ground-truth $\varphi(y)$. Our loss $S_{\mathcal{C}}^{\Psi}$ is a convex and **structurally informed** middle ground between $\ell_{\mathcal{C}}^{\Psi}$ and $S_{\mathrm{sq}}$.

How to choose $\mathcal{C}$? The smallest convex set $\mathcal{C}$ such that $\varphi(\mathcal{Y}) \subseteq \mathcal{C}$ is the **convex hull** of $\varphi(\mathcal{Y})$
$$\mathcal{M} \coloneqq \mathrm{conv}(\varphi(\mathcal{Y})) \coloneqq \{\mathbb{E}_{Y \sim q}[\varphi(Y)] : q \in \triangle^{|\mathcal{Y}|}\} \subseteq \Theta. \tag{8}$$
When $\varphi(y)$ is a zero-one encoding of the parts of $y$, $\mathcal{M}$ is also known as the **marginal polytope** [54], since any point inside it can be interpreted as some marginal distribution over parts of the structures. The loss $S_{\mathcal{C}}^{\Psi}$ with $\mathcal{C} = \mathcal{M}$ and $\Psi(u) = \frac{1}{2}\|u\|_2^2$ is exactly the sparseMAP loss proposed in [37]. More generally, we can use any **superset** $\mathcal{C}'$ of $\mathcal{M}$, with potentially cheaper-to-compute projections. For instance, when $\varphi(y)$ uses a zero-one encoding, the marginal polytope is always contained in the **unit cube**, i.e., $\mathcal{M} \subseteq [0, 1]^p$, whose projection is very cheap to compute. We show in our experiments that even just using the unit cube typically improves over the squared loss. However, an advantage of using $\mathcal{C} = \mathcal{M}$ is that $P_{\mathcal{M}}^{\Psi}(\theta)$ produces a convex combination of structures, i.e., an expectation.

**Smoothness.** The well-known equivalence between strong convexity of a function and the smoothness of its Fenchel conjugate implies that the following three statements are all equivalent:

- $\Psi$ is $\frac{1}{\beta}$-strongly convex w.r.t. a norm $\|\cdot\|$ over $\mathcal{C}$,

- $P_{\mathcal{C}}^{\Psi}$ is $\beta$-Lipschitz continuous w.r.t. the dual norm $\|\cdot\|_*$ over $\mathbb{R}^p$,

- $S_{\mathcal{C}}^{\Psi}$ is $\beta$-smooth in its first argument w.r.t. $\|\cdot\|_*$ over $\mathbb{R}^p$.

With the Euclidean geometry, since $\Psi(u) = \frac{1}{2}\|u\|_2^2$ is 1-strongly-convex over $\mathbb{R}^p$ w.r.t. $\|\cdot\|_2$, we have that $S_{\mathcal{C}}^{\Psi}$ is 1-smooth w.r.t. $\|\cdot\|_2$ **regardless** of $\mathcal{C}$. With the KL geometry, the situation is different. The fact that $\Psi(u) = \langle u, \log u \rangle$ is 1-strongly convex w.r.t. $\|\cdot\|_1$ over $\mathcal{C} = \triangle^p$ is well-known (this is Pinsker's inequality). The next proposition, proved in §C.1, shows that this straightforwardly extends to any bounded $\mathcal{C}$ and that the strong convexity constant is **inversely proportional** to the size of $\mathcal{C}$.

---

**Proposition 2** *Strong convexity of $\Psi(u) = \langle u, \log u \rangle$ over a bounded set*

*Let $\mathcal{C} \subseteq \mathbb{R}_+^d$ and $\beta \coloneqq \sup_{u \in \mathcal{C}} \|u\|_1$. Then, $\Psi$ is $\frac{1}{\beta}$-strongly convex w.r.t. $\|\cdot\|_1$ over $\mathcal{C}$.*

---

This implies that $S_{\mathcal{C}}^{\Psi}$ is $\beta$-smooth w.r.t. $\|\cdot\|_\infty$. Since smaller $\beta$ is smoother, this is another argument for **preferring smaller sets** $\mathcal{C}$. With the best choice of $\mathcal{C} = \mathcal{M}$, we obtain $\beta = \sup_{y \in \mathcal{Y}} \|\varphi(y)\|_1$.

**Computation.** Assuming $\mathcal{C}$ is compact (closed and bounded), the Euclidean projection can always be computed using Frank-Wolfe or active-set algorithms, provided access to a linear maximization oracle $\mathrm{LMO}_{\mathcal{C}}(v) \coloneqq \mathrm{argmax}_{u \in \mathcal{C}} \langle u, v \rangle$. Note that in the case $\mathcal{C} = \mathcal{M}$, assuming that $\varphi$ is injective, meaning that is has a left inverse, MAP inference reduces to an LMO, since $\mathrm{MAP}(\theta) = \varphi^{-1}(\mathrm{LMO}_{\mathcal{M}}(\theta))$ (the LMO can be viewed as a linear program, whose solutions always hit a vertex $\varphi(y)$ of $\mathcal{M}$). The KL projection is more problematic but Frank-Wolfe variants have been proposed [7, 27]. In the next section, we focus on examples of sets for which an efficient **dedicated** projection oracle is available.

## 4   Examples of convex polytopes and corresponding projections

**Probability simplex.** For **multiclass classification**, we set $\mathcal{Y} = [k]$, where $k$ is the number of classes. With $\varphi(y) = \mathbf{e}_y$, the one-hot encoding of $y$, MAP inference (2) becomes $\mathrm{MAP}(\theta) = \mathrm{argmax}_{i \in [k]} \theta_k$. The marginal polytope defined in (8) is now $\mathcal{M} = \triangle^k$, the probability simplex. The Euclidean and KL projections onto $\mathcal{C} = \mathcal{M}$ then correspond to the sparsemax [32] and softmax transformations. We therefore recover the sparsemax and logistic losses as natural special cases of $S_{\mathcal{C}}^{\Psi}$. Note that, although the CRF loss [29] also comprises the logistic loss as a special case, it no longer coincides with our loss in the structured case.

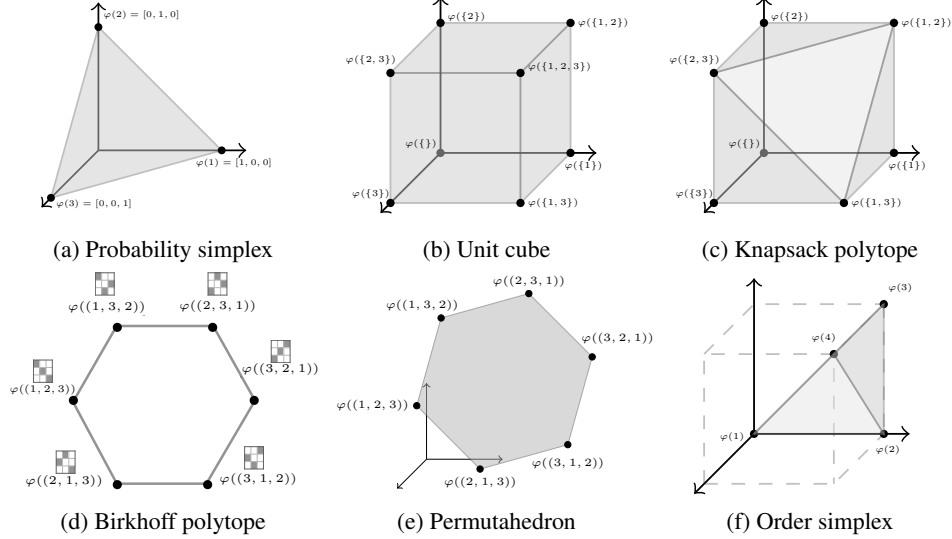

Figure 2: Examples of convex polytopes.

**Unit cube.** For **multilabel classification**, we choose $\mathcal{Y} = 2^{[k]}$, the powerset of $[k]$. Let us set $\varphi(y) = \sum_{i=1}^{|y|} \mathbf{e}_{y_i} \in \{0,1\}^k$, the label indicator vector of $y$ (i.e., $\varphi(y)_i = 1$ if $i \in y$ and 0 otherwise). MAP inference corresponds to predicting each label independently. More precisely, for each label $i \in [k]$, if $\theta_i > 0$ we predict $i$, otherwise we do not. The marginal polytope is now $\mathcal{M} = [0,1]^k$, the unit cube. Each vertex is in bijection with one possible subset of $[k]$. The Euclidean projection of $\theta$ onto $\mathcal{M}$ is equal to a coordinate-wise clipping of $\theta$, i.e., $\max(\min(\theta_i, 1), 0)$ for all $i \in [k]$. The KL projection is equal to $\min(1, e^{\theta_i - 1})$ for all $i \in [k]$. More generally, whenever $\varphi$ for the task at hand uses a 0-1 encoding, we can use the unit cube as superset with computationally cheap projection.

**Knapsack polytope.** We now set $\mathcal{Y} = \{y \in 2^{[k]} : l \le |y| \le u\}$, the subsets of $[k]$ of bounded size. We assume $0 \le l \le u \le k$. This is useful for **multilabel classification** with known lower bound $l \in \mathbb{N}$ and upper bound $u \in \mathbb{N}$ on the number of labels per sample. Setting again $\varphi(y) = \sum_{i=1}^{|y|} \mathbf{e}_{y_i} \in \{0,1\}^k$, MAP inference is equivalent to the integer linear program $\text{argmax}_{\varphi(y) \in \{0,1\}^k} \langle \theta, \varphi(y) \rangle$ s.t. $l \le \langle \varphi(y), \mathbf{1} \rangle \le u$. Let $\pi$ be a permutation sorting $\theta$ in descending order. An optimal solution is

$$\varphi(y)_i = \begin{cases} 1 & \text{if } l > 0 \text{ and } i \in \{\pi_1, \ldots, \pi_l\}, \\ 1 & \text{else if } i \in \{\pi_1, \ldots, \pi_u\} \text{ and } \theta_i > 0, \\ 0 & \text{else.} \end{cases}$$

The marginal polytope is an instance of knapsack polytope [2]. It is equal to $\mathcal{M} = \{\mu \in [0,1]^k : l \le \langle \mu, \mathbf{1} \rangle \le u\}$ and is illustrated in Figure 2c with $k = 3$, $l = 0$ and $u = 2$ (i.e., we keep all elements of $2^{[3]}$ except $\{1, 2, 3\}$). The next proposition, proved in §C.2, shows how to efficiently project on $\mathcal{M}$.

> **Proposition 3** *Efficient Euclidean and KL projections on $\mathcal{M}$*
>
> - *Let $\nu$ be the projection of $\nabla\Psi^*(\theta)$ onto the unit cube (cf. "unit cube" paragraph).*
> - *If $l \le \langle \nu, \mathbf{1} \rangle \le u$, then $\nu$ is optimal.*
> - *Otherwise, return the projection of $\nabla\Psi^*(\theta)$ onto $\{\mu \in [0,1]^k : \langle \mu, \mathbf{1} \rangle = m\}$, where $m = u$ if $\langle \nu, \mathbf{1} \rangle > u$ and $m = l$ otherwise.*

The total cost is $O(k)$ in the Euclidean case and $O(k \log k)$ in the KL case (cf. §C.2 for details).

**Birkhoff polytope.** We view **ranking** as a structured prediction problem and let $\mathcal{Y}$ be the set of permutations $\pi$ of $[k]$. Setting $\varphi(\pi) \in \{0,1\}^{k \times k}$ as the permutation matrix associated with $\pi$, MAP inference becomes the linear assignment problem $\text{MAP}(\theta) = \text{argmax}_{\pi \in \mathcal{Y}} \sum_{i=1}^{k} \theta_{i,\pi_i}$ and can be computed exactly using the Hungarian algorithm [28]. The marginal polytope $\mathcal{M}$ becomes the

Birkhoff polytope [10], the set of doubly stochastic matrices

$$\mathcal{M} = \{P \in \mathbb{R}^{k \times k} \colon P^\top \mathbf{1} = 1, P\mathbf{1} = 1, 0 \le P \le 1\}.$$

Noticeably, marginal inference is known to be #P-complete [53, 50, §3.5], since it corresponds to computing a matrix permanent. In contrast, the KL projection on the Birkhoff polytope can be computed using the Sinkhorn algorithm [47, 18]. The Euclidean projection can be computed using Dykstra's algorithm [19] or dual approaches [13]. For both projections, the cost of obtaining an $\epsilon$-precise solution is $O(k^2/\epsilon)$. To obtain cheaper projections, we can also use [13, 38] the set $\triangle^{k \times k}$ of row-stochastic matrices, a strict superset of the Birkhoff polytope and strict subset of the unit cube

$$[0,1]^{k \times k} \supset \triangle^{k \times k} := \triangle^k \times \cdots \times \triangle^k = \{P \in \mathbb{R}^{k \times k} \colon P^\top \mathbf{1} = 1, 0 \le P \le 1\} \supset \mathcal{M}.$$

Projections onto $\triangle^{k \times k}$ reduce to $k$ row-wise projections onto $\triangle^k$, for a *worst-case* total cost of $O(k^2 \log k)$ in the Euclidean case and $O(k^2)$ in the KL case.

**Permutahedron.** We again consider **ranking** and let $\mathcal{Y}$ be the set of permutations $\pi$ of $[k]$ but use a different encoding. This time, we define $\varphi(\pi) = (w_{\pi_1}, \ldots, w_{\pi_k}) \in \mathbb{R}^k$, where $w \in \mathbb{R}^k$ is a prescribed vector of weights, which without loss of generality, we assume sorted in descending order. MAP inference becomes $\mathrm{MAP}(\theta) = \mathrm{argmax}_{\pi \in \mathcal{Y}} \sum_{i=1}^k \theta_i w_{\pi_i} = \sum_{i=1}^k \theta_{\pi_i^{-1}} w_i$, where $\pi^{-1}$ denotes the inverse permutation of $\pi$. The MAP solution is thus the inverse of the permutation sorting $\theta$ in descending order, and can be computed in $O(k \log k)$ time. When $w = (k, \ldots, 1)$, which we use in our experiments, $\mathcal{M}$ is known as the permutahedron. For arbitrary $w$, we follow [30] and call $\mathcal{M}$ the permutahedron induced by $w$. Its vertices correspond to the permutations of $w$. Importantly, the Euclidean projection onto $\mathcal{M}$ reduces to sorting, which takes $O(k \log k)$, followed by isotonic regression, which takes $O(k)$ [57, 36]. Bregman projections reduce to isotonic *optimization* [30].

**Order simplex.** We again set $\mathcal{Y} = [k]$ but now consider the **ordinal regression** setting, where there is an intrinsic order $1 \prec \cdots \prec k$. We need to use an encoding $\varphi$ that takes into account that order. Inspired by the all-threshold method [42, 38], we set $\varphi(y) = \sum_{1 \le i < y \le k} \mathbf{e}_i \in \mathbb{R}^{k-1}$. For instance, with $k = 4$, we have $\varphi(1) = [0,0,0]$, $\varphi(2) = [1,0,0]$, $\varphi(3) = [1,1,0]$ and $\varphi(4) = [1,1,1]$. This encoding is also motivated by the fact that it enables consistency w.r.t. the absolute loss (§A). As proved in §C.3, with that encoding, the marginal polytope becomes the order simplex [22].

---

**Proposition 4** *Vertices of the order simplex*

$$\mathcal{M} = \mathrm{conv}(0, e_1, e_1 + e_2, \ldots, e_1 + \cdots + e_{k-1}) = \{\mu \in \mathbb{R}^{k-1} \colon 1 \ge \mu_1 \ge \mu_2 \ge \cdots \ge \mu_{k-1} \ge 0\}$$

---

Note that without the upper bound on $\mu_1$, the resulting set is known as monotone nonnegative cone [14]. The scores $(\langle \theta, \varphi(y) \rangle)_{y=1}^k$ can be calculated using a cumulated sum in $O(k)$ time and therefore so do MAP and marginal inferences. The Euclidean projection is equivalent to isotonic regression with lower and upper bounds, which can be computed in $O(k)$ time [9].

# 5 Consistency analysis of projection-based losses

We now study the consistency of $S_\mathcal{C}^\Psi$ as a proxy for a possibly non-convex target loss $L \colon \mathcal{Y} \times \mathcal{Y} \to \mathbb{R}_+$.

**Affine decomposition.** We assume that the target loss $L$ satisfies the decomposition

$$L(\widehat{y}, y) = \langle \varphi(\widehat{y}), V\varphi(y) + b \rangle + c(y). \tag{9}$$

This is a slight generalization of the decomposition of [15], where we used an affine map $u \mapsto Vu + b$ instead of a linear one and where we added the term $c \colon \mathcal{Y} \to \mathbb{R}$, which is independent of $\widehat{y}$. This modification allows us to express certain losses $L$ using a zero-one encoding for $\varphi$ instead of a signed encoding [38]. The latter is problematic when using KL projections and does not lead to sparse solutions with Euclidean projections. Examples of target losses satisfying (9) are discussed in §A.

**Calibrated decoding.** A drawback of the classical inference pipeline (1) with decoder $d = \mathrm{MAP}$ is that it is oblivious to the target loss $L$. In this paper, we propose to use instead

$$x \in \mathcal{X} \xrightarrow[\text{model}]{g} \theta \in \Theta = \mathbb{R}^p \xrightarrow[\text{projection}]{P_\mathcal{C}^\Psi} u \in \mathcal{C} \xrightarrow[\text{calibrated decoding}]{\widehat{y}_L} \widehat{y} \in \mathcal{Y}, \tag{10}$$

where we define the decoding calibrated for the loss $L$ by

$$\widehat{y}_L(u) := \operatorname*{argmin}_{y' \in \mathcal{Y}} \langle \varphi(y'), Vu + b \rangle = \mathrm{MAP}(-Vu - b). \tag{11}$$

Under the decomposition (9), calibrated decoding therefore reduces to MAP inference with pre-processed input. It is a "rounding" to $\mathcal{Y}$ of the projection $u = P_{\mathcal{C}}^{\Psi}(\theta) \in \mathcal{C}$, that takes into account the loss $L$. Recently, [15, 26, 39, 31] used similar calibrated decoding in conjunction with a squared loss (i.e., without an intermediate layer) and [38] used it with a CRF loss (with marginal inference as intermediate layer). To our knowledge, we are the first to use a **projection layer** (in the Euclidean or KL senses) as an intermediate step.

**Calibrating target and surrogate excess risks.** Given a (typically unknown) joint distribution $\rho \in \triangle(\mathcal{X} \times \mathcal{Y})$, let us define the target risk of $f \colon \mathcal{X} \to \mathcal{Y}$ and the surrogate risk of $g \colon \mathcal{X} \to \Theta$ by

$$\mathcal{L}(f) := \mathbb{E}_{(X,Y) \sim \rho} \, L(f(X), Y) \quad \text{and} \quad \mathcal{S}_{\mathcal{C}}^{\Psi}(g) := \mathbb{E}_{(X,Y) \sim \rho} \, S_{\mathcal{C}}^{\Psi}(g(X), Y).$$

The quality of estimators $f$ and $g$ is measured in terms of the *excess* of risks

$$\delta\mathcal{L}(f) := \mathcal{L}(f) - \inf_{f' \colon \mathcal{X} \to \mathcal{Y}} \mathcal{L}(f') \quad \text{and} \quad \delta\mathcal{S}_{\mathcal{C}}^{\Psi}(g) := \mathcal{S}_{\mathcal{C}}^{\Psi}(g) - \inf_{g' \colon \mathcal{X} \to \Theta} \mathcal{S}_{\mathcal{C}}^{\Psi}(g').$$

The following proposition shows that $\delta\mathcal{L}(f)$ and $\delta\mathcal{S}_{\mathcal{C}}^{\Psi}(g)$ are calibrated when using our proposed inference pipeline (10), i.e., when $f = \widehat{y}_L \circ P_{\mathcal{C}}^{\Psi} \circ g$.

---

**Proposition 5** *Calibration of target and surrogate excess risks*

*Let $S_{\mathcal{C}}^{\Psi}(\theta, y)$ and $L(\widehat{y}, y)$ be defined as in (6) and (9), respectively. Assume $\Psi$ is $\frac{1}{\beta}$-strongly convex w.r.t. $\| \cdot \|$ over $\mathcal{C}$, Legendre-type, and $\mathcal{C}$ is a closed convex set such that $\varphi(\mathcal{Y}) \subseteq \mathcal{C} \subseteq \mathrm{dom}(\Psi)$. Let $\sigma := \sup_{\widehat{y} \in \mathcal{Y}} \|V^{\top} \varphi(\widehat{y})\|_*$, where $\| \cdot \|_*$ is the dual norm of $\| \cdot \|$. Then,*

$$\forall g \colon \mathcal{X} \to \Theta : \quad \frac{\delta\mathcal{L}(\widehat{y}_L \circ P_{\mathcal{C}}^{\Psi} \circ g)^2}{8\beta\sigma^2} \leq \delta\mathcal{S}_{\mathcal{C}}^{\Psi}(g).$$

---

The proof, given in §C.4, is based on the calibration function framework of [40] and extends a recent analysis [38] to projection-based losses. Our proof covers Euclidean projection losses, not covered by the previous analysis. Proposition 5 implies Fisher consistency, i.e., $\mathcal{L}(\widehat{y}_L \circ P_{\mathcal{C}}^{\Psi} \circ g^{\star}) = \mathcal{L}(f^{\star})$, where $f^{\star} := \operatorname{argmin}_{f \colon \mathcal{X} \to \mathcal{Y}} \mathcal{L}(f)$ and $g^{\star} := \operatorname{argmin}_{g \colon \mathcal{X} \to \Theta} \mathcal{S}_{\mathcal{C}}^{\Psi}(g)$. Consequently, any optimization algorithm converging to $g^{\star}$ will also recover an optimal estimator $\widehat{y}_L \circ P_{\mathcal{C}}^{\Psi} \circ g^{\star}$ of $\mathcal{L}$. Combined with Propositions 1 and 2, Proposition 5 suggests a **trade-off** between computational cost and statistical estimation, larger sets $\mathcal{C}$ enjoying cheaper-to-compute projections but leading to slower rates.

# 6 Experimental results

We present in this section our empirical findings on three tasks: label ranking, ordinal regression and multilabel classification. In all cases, we use a linear model $\theta = g(x) := Wx$ and solve $\frac{1}{n}\sum_{i=1}^{n} S_{\mathcal{C}}^{\Psi}(Wx_i, y_i) + \frac{\lambda}{2}\|W\|_F^2$ by L-BFGS, choosing $\lambda$ against the validation set. A Python implementation is available at `https://github.com/mblondel/projection-losses`.

**Label ranking.** We consider the label ranking setting where supervision is given as full rankings (e.g., $2 \succ 1 \succ 3 \succ 4$) rather than as label relevance scores. Note that the exact CRF loss is intractable for this task. We use the same six public datasets as in [26]. We compare different convex sets for the projection $P_{\mathcal{C}}^{\Psi}$ and the decoding $\widehat{y}_L$. For the Euclidean and KL projections onto the Birkhoff polytope, we solve the semi-dual formulation [13] by L-BFGS. We report the mean Hamming loss, for which our loss is consistent, between the ground-truth and predicted permutation matrices in the test set. Results are shown in Table 1 and Table 2. We summarize our findings below.

- For decoding, using $[0,1]^{k \times k}$ or $\triangle^{k \times k}$ instead of the Birkhoff polytope considerably degrades accuracy. This is not surprising, as these choices do not produce valid permutation matrices.

- Using a squared loss $\frac{1}{2}\|\varphi(y) - \theta\|^2$ ($\mathcal{C} = \mathbb{R}^{k \times k}$, no projection) works relatively well when combined with permutation decoding. Using supersets of the Birkhoff polytope as projection set $\mathcal{C}$, such as $[0,1]^{k \times k}$ or $\triangle^{k \times k}$, improves accuracy substantially. However, the best accuracy is obtained when using the Birkhoff polytope for **both** projections and decoding.

Table 1: Hamming loss (lower is better) for label ranking with Euclidean projections. The first line indicates the projection set $\mathcal{C}$ used in (5). The second line indicates the decoding set used in (11). Using the Birkhoff polytope for **both** projections and decoding achieves the best accuracy.

| Projection | $[0,1]^{k\times k}$ | $\triangle^{k\times k}$ | $\mathbb{R}^{k\times k}$ | $[0,1]^{k\times k}$ | $\triangle^{k\times k}$ | $\mathcal{M}$ |
| Decoding | $[0,1]^{k\times k}$ | $\triangle^{k\times k}$ | $\mathcal{M}$ | $\mathcal{M}$ | $\mathcal{M}$ | $\mathcal{M}$ |
|---|---|---|---|---|---|---|
| Authorship | 12.83 | 5.62 | 5.70 | 5.18 | 5.70 | **5.10** |
| Glass | 24.35 | 5.43 | 7.11 | 5.68 | 5.04 | **4.65** |
| Iris | 27.78 | 10.37 | 19.26 | 4.44 | **1.48** | 2.96 |
| Vehicle | 26.36 | 7.43 | 9.04 | 7.57 | 6.99 | **5.88** |
| Vowel | 43.71 | 9.65 | 10.57 | 9.56 | 9.18 | **8.76** |
| Wine | 10.19 | 1.85 | **1.23** | 1.85 | 1.85 | 1.85 |

$\mathcal{M}$: Birkhoff polytope

| Projection | $\triangle^{k\times k}$ | $\mathbb{R}_+^{k\times k}$ | $[0,1]^{k\times k}$ | $\triangle^{k\times k}$ | $\mathcal{M}$ |
| Decoding | $\triangle^{k\times k}$ | $\mathcal{M}$ | $\mathcal{M}$ | $\mathcal{M}$ | $\mathcal{M}$ |
|---|---|---|---|---|---|
| Authorship | 5.84 | **5.10** | 5.62 | 5.84 | **5.10** |
| Glass | 5.43 | 5.81 | 5.94 | 5.68 | **4.65** |
| Iris | 11.11 | 18.52 | 4.44 | **1.48** | 2.96 |
| Vehicle | 7.57 | 8.46 | 7.43 | 7.21 | **6.25** |
| Vowel | 9.50 | 9.40 | 9.42 | 9.28 | **9.17** |
| Wine | 4.32 | **1.85** | **1.85** | **1.85** | **1.85** |

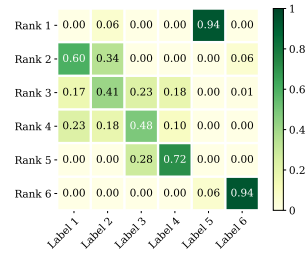

Table 2: Same as Table 1 but with KL projections instead.

Figure 3: Example of soft permutation matrix.

- The losses derived from Euclidean and KL projections perform similarly. This is informative, as algorithms for Euclidean projections onto various sets are more widely available.

Beyond accuracy improvements, the projection $\mu = P_\mathcal{M}^\Psi(Wx)$ is useful to visualize soft permutation matrices predicted by the model, an advantage lost when using supersets of the Birkhoff polytope.

**Ordinal regression.** We compared classical ridge regression to our order simplex based loss on sixteen publicly-available datasets [23]. For evaluation, we use mean absolute error (MAE), for which our loss is consistent when suitably setting $V$ and $b$ (cf. §A). We find that ridge regression performs the worst with an average MAE of $0.72$. Combining a squared loss $\frac{1}{2}\|\varphi(y) - \theta\|^2$ (no projection) with order simplex decoding at prediction time improves the MAE to $0.47$. Using a projection on the unit cube, a superset of the order simplex, further improves the MAE to $0.45$. Finally, using the Euclidean projection onto the order simplex achieves the best MAE of $0.43$, confirming that using the order simplex for both projections and decoding works better. Detailed results are reported in Table 4.

**Multilabel classification.** We compared losses derived from the unit cube and the knapsack polytope on the same seven datasets as in [32, 11]. We set the lower bound $l$ to $0$ and the upper-bound $u$ to $\lceil \mathbb{E}[|Y|] + \sqrt{\mathbb{V}[|Y|]} \rceil$, where $\mathbb{E}$ and $\mathbb{V}$ are computed over the training set. Although the unit cube is a strong baseline, we find that the knapsack polytope improves $F_1$ score on some datasets, especially with few labels per sample ("birds", "emotions", "scene"). Results are reported in Tables 6 and 7.

## 7   Conclusion

We proposed in this paper a general framework for deriving a smooth and convex loss function from the projection onto a convex set, bringing a computational geometry perspective to structured prediction. We discussed several examples of polytopes with efficient Euclidean or KL projection, making our losses useful for a variety of structured tasks. Our theoretical and empirical results suggest that the marginal polytope is the convex set of choice when the projection onto it is affordable. When not, our framework allows to use any superset with cheaper-to-compute projection.

**Acknowledgments**

We thank Vlad Niculae for suggesting the knapsack polytope for multilabel classification, and Tomoharu Iwata for suggesting to add a lower bound on the number of labels. We also thank Naoki Marumo for numerous fruitful discussions.

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
