[Supplementary Material]

# Appendix

## A    Examples of decomposable target losses

For more generality, following [38], we consider losses $L\colon \mathcal{O} \times \mathcal{Y} \to \mathbb{R}_+$, where $\mathcal{O}$ is the output space and $\mathcal{Y}$ is the ground-truth space. Typically, $\mathcal{O} = \mathcal{Y}$ but we give examples below where $\mathcal{O} \neq \mathcal{Y}$. Our affine decomposition (9) now becomes

$$L(\widehat{y}, y) = \langle \psi(\widehat{y}), V\varphi(y) + b \rangle + c(y), \tag{12}$$

where $\psi\colon \mathcal{O} \to \mathbb{R}^p$. We give examples below of possibly non-convex losses $L$ satisfiying decomposition (12). When not mentioned explicitly, we set $V = I$, $b = \mathbf{0}$ and $c(y) = 0$. For more examples of decomposable losses, see also [15, 39, 38].

**General loss.**    As noted in [15], any loss $L\colon \mathcal{O} \times \mathcal{Y} \to \mathbb{R}_+$ can always be written as (12) if $\mathcal{O}$ and $\mathcal{Y}$ are finite sets. Indeed, it suffices to set $\psi(y) = \varphi(y) = \mathbf{e}_y$ and to define $V$ as the $|\mathcal{O}| \times |\mathcal{Y}|$ loss matrix, i.e., $V_{\widehat{y},y} = L(\widehat{y}, y)$ for all $\widehat{y} \in \mathcal{O}$ and $y \in \mathcal{Y}$. This, however, ignores structural information, essential for large output spaces encountered in structured prediction.

**Zero-one loss for multiclass classification.**    Let $\mathcal{O} = \mathcal{Y} = [k]$. The 0-1 loss $L(\widehat{y}, y) = \mathbf{1}[\widehat{y} \neq y]$ can be written as (12) if we set $\psi(y) = \varphi(y) = \mathbf{e}_y$ and $V = 1 - I_{k \times k}$, i.e., $V$ is the 0-1 cost matrix.

**Hamming loss for multilabel classification.**    Let $\mathcal{O} = \mathcal{Y} = 2^{[k]}$ and $\varphi(y) = \sum_{i=1}^{|y|} \mathbf{e}_{y_i}$. Then,

$$L(\widehat{y}, y) = \sum_{i=1}^{k} \mathbf{1}[\widehat{y}_i \neq y_i] = \langle \varphi(\widehat{y}), \mathbf{1} \rangle + \langle \varphi(y), \mathbf{1} \rangle - 2\langle \varphi(\widehat{y}), \varphi(y) \rangle.$$

This can be written as (12) with $V = -2I$, $b = \mathbf{1}$ and $c(y) = \langle \varphi(y), \mathbf{1} \rangle$.

**Hamming loss for ranking.**    Let $\mathcal{O} = \mathcal{Y}$ be the set of permutations of $[k]$. If $\psi(y) = \varphi(y)$ is the permutation matrix associated with permutation $y$, the Hamming loss is $L(\widehat{y}, y) = \sum_{i=1}^{k} \mathbf{1}[\widehat{y}_i \neq y_i] = k - \langle \varphi(\widehat{y}), \varphi(y) \rangle$. It can thus be written as (12) with $V = -I_{k \times k}$ and $c(y) = k$.

**Normalized discounted cumulative gain (NDCG).**    Let $\mathcal{O}$ be the permutations of $[m]$ and $\mathcal{Y} = [k]^m$ be the relevance scores of $m$ documents. The NDCG loss is $L(\pi, y) = 1 - \frac{1}{N(y)} \sum_{i=1}^{m} y_i w_{\pi_i}$, where $N(y) = \max_{\pi \in \mathcal{O}} \sum_{i=1}^{m} y_i w_{\pi_i}$ is a normalization constant. Inspired by [38], we can thus write $L$ as (12) by defining $\psi(\pi)$ as the permutation of $w$ according to $\pi$, $\varphi(y) = y/N(y)$, $V = -I$ and $c(y) = 1$. This shows the importance of learning to predict normalized relevance scores, as also noted in [45]. For this reason, we suggest using $\mathcal{C} = \mathbb{R}_+^m$. Decoding reduces to linear maximization over the permutahedron.

**Precision at $k$ for ranking.**    Let $\mathcal{O}$ be the permutations of $[m]$ and $\mathcal{Y} = \{0,1\}^m$ be binary relevance scores. Precision at $k$ corresponds to the number of relevant results (e.g., labels or documents) in the top $k$ results. The corresponding loss can be defined by $L(\pi, y) = 1 - \frac{1}{k} \sum_{i=1}^{m} y_i w_{\pi_i}$, where $w_1 = \cdots = w_k = 1$ and $w_{k+1} = \cdots = w_n = 0$. If the number of positive labels $|y| = \sum_{i=1}^{m} y_i$ is less than $k$, we replace $k$ with $|y|$. Similarly as for NDCG, we can therefore write $L$ as (12). Again, decoding reduces to linear maximization over the permutahedron.

**Absolute loss for ordinal regression.**    Let $\mathcal{O} = \mathcal{Y} = [k]$ and $\psi(y) = \varphi(y) = \sum_{i<y} \mathbf{e}_i$. Then,

$$L(\widehat{y}, y) = |\widehat{y} - y| = \langle \varphi(\widehat{y}), \mathbf{1} \rangle + \langle \varphi(y), \mathbf{1} \rangle - 2\langle \varphi(\widehat{y}), \varphi(y) \rangle.$$

This can be written as (12) with $V = -2I$, $b = \mathbf{1}$ and $c(y) = \langle \varphi(y), \mathbf{1} \rangle$. A similar loss decomposition is derived in [38] in the case of a signed encoding, instead of the zero-one encoding we use. However, the signed encoding is problematic when using KL projections and does not lead to sparse projections when using Euclidean projections.

# B   Experiment details and additional empirical results

We discuss in this section our experimental setup and additional empirical results.

For all datasets, we normalized samples to have zero mean unit variance. We use the train-test split from the dataset when provided. When not, we use 80% for training data and 20% for test data. We hold out 25% of the training data for hyperparameter validation purposes. For the regularization hyper-parameter $\lambda$, we used ten log-spaced values between $10^{-4}$ and $10^4$. Once we select the best hyperparameter, we refit the model on the entire training set. We ran all experiments on a machine with Intel(R) Xeon(R) CPU with 2.90GHz and 4GB of RAM.

In all experiments, we use publicly-available datasets:

- `https://github.com/akorba/Structured_Approach_Label_Ranking`
- `http://www.uco.es/grupos/ayrna/ucobigfiles/datasets-orreview.zip`
- `http://mulan.sourceforge.net/datasets-mlc.html`
- `https://www.csie.ntu.edu.tw/~cjlin/libsvmtools/datasets/`

## B.1   Label ranking

In this section, we compare the Birkhoff and permutahedron polytopes for the same label ranking task as described in the main manuscript. With the Birkhoff polytope, $\theta = g(x) \in \mathbb{R}^{k \times k}$ can be interpreted as an affinity matrix between classes for the input $x$. With the permutahedron, $\theta = g(x) \in \mathbb{R}^k$ can be intepreted as vector, containing the score of each class for input $x$. Therefore, the two polytopes have different expressive power. For the model $g(x)$, we compare $g(x) = Wx$, where $W$ is a matrix or linear map of proper shape, and a polynomial model $g(x) = \sum_{i=1}^n w_i \kappa(x, x_i)$, where $w \in \mathbb{R}^n$, $\kappa(x, x') \coloneqq (\langle x, x' \rangle + 1)^D$ and $D$ is the polynomial degree. For the Euclidean projection onto the permutahedron, we use the isotonic regression solver from scikit-learn [43].

Our results, shown in Table 3, indicate that in the case of a linear model, the Birkhoff polytope outperforms the permutahedron by a large margin. Using a polynomial model closes the gap between the two, but the model based on the Birkhoff polytope is still slightly better.

Table 3: Test Hamming loss comparison when using the Birkhoff and permutahedron polytopes.

| Projection | $\mathcal{B}$ | $\mathcal{P}$ | $\mathcal{P}$ | $\mathcal{P}$ |
| Decoding | $\mathcal{B}$ | $\mathcal{P}$ | $\mathcal{P}$ | $\mathcal{P}$ |
| Model | Linear | Linear | Poly ($D=2$) | Poly ($D=3$) |
|---|---|---|---|---|
| Authorship | **5.10** | 10.06 | 10.50 | 8.59 |
| Glass | **4.65** | 7.49 | 7.10 | 8.14 |
| Iris | **2.96** | 27.41 | 20.00 | 5.93 |
| Vehicle | **5.88** | 11.62 | 8.30 | 9.26 |
| Vowel | **8.76** | 14.35 | 11.74 | 10.21 |
| Wine | **1.85** | 8.02 | 3.08 | 6.79 |

$\mathcal{B}$: Birkhoff polytope, $\mathcal{P}$: permutahedron

## B.2 Ordinal regression

In this section, we present our detailed results on ordinal regression. Table 4 below shows the results for each dataset. For context, the first column indicates a simple baseline in which we always predict the median label calculated on the train set. The second column indicates classical ridge regression where we used rounding to the closest integer as decoding. Using the order simplex for both projections and decoding achieves the best MAE on average.

Table 4: Mean absolute error (MAE) of our losses with Euclidean projections.

| Projection Decoding | Baseline | $\mathbb{R}$ Round | $\mathbb{R}^{k-1}$ $\mathcal{M}$ | $[0,1]^{k-1}$ $\mathcal{M}$ | $\mathcal{M}$ $\mathcal{M}$ |
|---|---|---|---|---|---|
| ERA | 1.61 | 1.24 | **1.19** | **1.19** | **1.19** |
| ESL | 1.12 | 0.34 | 0.37 | **0.30** | **0.30** |
| LEV | 0.71 | **0.41** | 0.48 | 0.46 | 0.46 |
| SWD | 0.63 | 0.47 | **0.41** | 0.42 | 0.42 |
| Automobile | 1.02 | 0.50 | 0.54 | 0.54 | **0.48** |
| Balance-scale | 0.91 | 0.29 | **0.10** | **0.10** | **0.10** |
| Car | 0.41 | 0.32 | 0.23 | **0.17** | **0.17** |
| Contact-lenses | 0.5 | 0.50 | 0.33 | **0.17** | **0.17** |
| Eucalyptus | 1.22 | 3.96 | **0.40** | 0.44 | 0.45 |
| Newthyroid | 0.29 | 0.09 | 0.11 | **0.02** | **0.02** |
| Pasture | 0.66 | 0.44 | **0.33** | **0.33** | **0.33** |
| Squash-stored | 0.54 | **0.38** | 0.54 | 0.62 | 0.62 |
| Squash-unstored | 0.54 | 0.46 | 0.31 | 0.31 | **0.15** |
| Tae | 0.69 | **0.66** | **0.66** | **0.66** | **0.66** |
| Toy | 0.97 | **0.96** | 0.97 | 0.97 | 0.97 |
| Winequality-red | 0.66 | 0.46 | **0.45** | 0.46 | 0.46 |
| Average MAE | 0.78 | 0.72 | 0.47 | 0.45 | 0.43 |
| Average rank | 4.75 | 2.9 | 2.1 | 1.6 | **1.5** |

$\mathcal{M}$: order simplex

## B.3 Multilabel classification

In this section, we show full empirical results for our multilabel experiment. Dataset statistics are summarized in Table 5. Empirical results are shown in Tables 6 and 7.

Table 5: Multilabel dataset statistics

| Dataset | Type | Train | Dev | Test | Features | Classes | Avg. labels |
|---|---|---|---|---|---|---|---|
| Birds | Audio | 134 | 45 | 172 | 260 | 19 | 1.96 |
| Cal500 | Music | 376 | 126 | 101 | 68 | 174 | 25.98 |
| Emotions | Music | 293 | 98 | 202 | 72 | 6 | 1.82 |
| Mediamill | Video | 22,353 | 7,451 | 12,373 | 120 | 101 | 4.54 |
| Scene | Images | 908 | 303 | 1,196 | 294 | 6 | 1.06 |
| SIAM TMC | Text | 16,139 | 5,380 | 7,077 | 30,438 | 22 | 2.22 |
| Yeast | Micro-array | 1,125 | 375 | 917 | 103 | 14 | 4.17 |

Table 6: Accuracy comparison on multilabel classification, using Euclidean projections.

| Projection | $[0,1]^k$ | $\mathbb{R}^k$ | $[0,1]^k$ | $\mathcal{M}$ |
|---|---|---|---|---|
| Decoding | $[0,1]^k$ | $\mathcal{M}$ | $\mathcal{M}$ | $\mathcal{M}$ |
| Birds | 90.21 | 90.61 | 90.61 | **90.70** |
| Cal500 | 85.78 | 85.76 | 85.78 | **85.80** |
| Emotions | **77.64** | 76.32 | 75.99 | 75.83 |
| Mediamill | **96.90** | 96.85 | **96.90** | 96.82 |
| Scene | 90.05 | 88.34 | 89.62 | **90.79** |
| TMC | **94.67** | 94.48 | 94.65 | **94.67** |
| Yeast | **79.75** | 79.70 | **79.75** | 79.71 |

$\mathcal{M}$: knapsack polytope

Table 7: $F_1$-score comparison on multilabel classification, using Euclidean projections.

| Projection | $[0,1]^k$ | $\mathbb{R}^k$ | $[0,1]^k$ | $\mathcal{M}$ |
|---|---|---|---|---|
| Decoding | $[0,1]^k$ | $\mathcal{M}$ | $\mathcal{M}$ | $\mathcal{M}$ |
| Birds | 38.87 | 37.75 | 39.21 | **39.43** |
| Cal500 | 34.62 | **35.86** | 34.63 | 34.61 |
| Emotions | 56.60 | 51.73 | 53.98 | **62.57** |
| Mediamill | **56.22** | 55.35 | **56.22** | 54.53 |
| Scene | 61.06 | 50.33 | 58.95 | **69.01** |
| TMC | **60.45** | 58.61 | 60.37 | 60.25 |
| Yeast | **60.24** | 60.20 | 60.23 | 60.06 |

$\mathcal{M}$: knapsack polytope

# C  Proofs

## C.1  Proof of strong convexity of Shannon negentropy (Proposition 2)

Let $\mathcal{C} \subseteq \mathbb{R}^d_+$ and $\Psi(u) = \langle u, \log u \rangle$. For all $u, v \in \mathcal{C}$ we have [14, §9.1.2]

$$\Psi(u) = \Psi(v) + \nabla\Psi(v)^\top(u-v) + \frac{1}{2}(u-v)^\top\nabla^2\Psi(w)(u-v),$$

for some $w \in \mathcal{C}$ in the line segment $[u, v]$, and where $\nabla\Psi(u) = \log u + 1$, $\nabla^2\Psi(u) = \text{diag}(u^{-1})$.
Recall that $\Psi$ is $\frac{1}{\beta}$-strongly convex over $\mathcal{C}$ w.r.t. $\|\cdot\|$ if for all $u, v \in \mathcal{C}$

$$\Psi(u) \geq \Psi(v) + \nabla\Psi(v)^\top(u-v) + \frac{1}{2\beta}\|u-v\|^2. \tag{13}$$

Therefore, letting $z = u - v$, it suffices to show that for all $u, v, w \in \mathcal{C}$

$$\beta z^\top\nabla^2\Psi(w)z \geq \|z\|_1^2. \tag{14}$$

Note that if there exists $w_i = 0$, then (14) clearly holds. Therefore we can focus on showing (14) for $w > 0$. This is indeed verified since by the Cauchy–Schwarz inequality

$$\|z\|_1^2 = \left(\sum_{i=1}^d \frac{|z_i|}{\sqrt{w_i}}\sqrt{w_i}\right)^2 \leq \sum_{i=1}^d \frac{z_i^2}{w_i}\sum_{i=1}^d w_i = z^\top\nabla^2\Psi(w)z\,\|w\|_1.$$

Therefore $\Psi$ is $\frac{1}{\beta}$-strongly convex over $\mathcal{C}$ w.r.t. $\|\cdot\|_1$, with $\beta = \sup_{w' \in \mathcal{C}}\|w'\|_1$.

## C.2  Projection onto the knapsack polytope (Proposition 3)

**Euclidean case.**  The Euclidean projection onto the knapsack polytope is

$$\underset{\mu \in \mathbb{R}^k}{\text{argmin}}\;\frac{1}{2}\|\mu - \theta\|^2 \quad \text{s.t.} \quad l \leq \langle\mu, \mathbf{1}\rangle \leq u, \quad 0 \leq \mu \leq 1.$$

The corresponding Lagrangian is

$$\mathcal{L} = \frac{1}{2}\|\mu - \theta\|^2 + \tau(\langle\mu, \mathbf{1}\rangle - u) + \eta(l - \langle\mu, \mathbf{1}\rangle) - \langle\xi, \mu\rangle + \langle\zeta, \mu - \mathbf{1}\rangle,$$

where the dual feasibility conditions are $\tau \geq 0$, $\eta \geq 0$, $\xi \in \mathbb{R}^k_+$ and $\zeta \in \mathbb{R}^k_+$. From the stationary conditions, the optimal $\mu$ should satisfy

$$\mu = \theta + (\eta - \tau)\mathbf{1} + \xi - \zeta.$$

From the complementary slackness conditions,

$$\tau(\langle\mu, \mathbf{1}\rangle - u) = 0$$
$$\eta(l - \langle\mu, \mathbf{1}\rangle) = 0$$
$$\zeta_i(\mu_i - 1) = 0 \quad \forall i \in [k]$$
$$\xi_i\mu_i = 0 \quad \forall i \in [k].$$

If $0 < \mu_i < 1$, then $\xi_i = \zeta_i = 0$ and $\mu_i = \theta_i - \tau + \eta$. If $\xi_i > 0$ then $\mu_i = 0$. If $\zeta_i > 0$ then $\mu_i = 1$. Altogether, we thus have for all $i \in [k]$

$$\mu_i = \text{clip}_{[0,1]}(\theta_i - \tau + \eta) := \min\{1, \max\{0, \theta_i - \tau + \eta\}\}.$$

Three cases can happen:

- If $\tau = \eta = 0$, the inequality $l \leq \langle\mu, \mathbf{1}\rangle \leq u$ is inactive. Therefore the projection $\text{clip}_{[0,1]}(\theta)$ onto the unit cube is optimal.
- If $\tau > 0$, then $\eta = 0$ and $\langle\mu, \mathbf{1}\rangle = u$ is active. This case happens when $\langle\text{clip}_{[0,1]}(\theta), \mathbf{1}\rangle > u$.
- If $\eta > 0$, then $\tau = 0$ and $l = \langle\mu, \mathbf{1}\rangle$ is active. This case happens when $\langle\text{clip}_{[0,1]}(\theta), \mathbf{1}\rangle < l$.

The second and third cases correspond to a projection onto $\{\mu \in \mathbb{R}^k : \langle\mu, \mathbf{1}\rangle = m, 0 \leq \mu \leq 1\}$, with $m = u$ or $m = l$. This projection can be computed in $O(k)$ time using Pardalos and Kovoor's algorithm [41]. See also [2, Appendix A] for pseudo code. Since that set is a special case of permutahedron with $w \in \mathbb{R}^k$ defined by $w_1 = \cdots = w_m = 1$ and $w_{m+1} = \cdots = w_k = 0$, we can also use the projection onto the permutahedron. The cost is $O(k \log k)$ for sorting $\theta$ and $O(k)$ for isotonic regression via the pool adjacent violators algorithm [30]. Yet another alternative is to search for $\tau$ solving $\sum_{i=1}^k \text{clip}_{[0,1]}(\theta_i - \tau) = m$ by bisection.

**KL case.** We want to solve (note that the non-negativity constraint on $\mu$ is vacuous)

$$\underset{\mu \in \mathbb{R}^k}{\operatorname{argmin}} \langle \mu, \log \mu \rangle - \langle \mu, \theta \rangle \quad \text{s.t.} \quad l \le \langle \mu, \mathbf{1} \rangle \le u, \quad 0 \le \mu \le 1.$$

As for the Euclidean projection, we consider three cases.

- If the projection $\nu = \min(1, e^{\theta-1})$ on the unit cube satisfies the constraints, $\nu$ is the optimal solution.
- If $\langle \nu, \mathbf{1} \rangle > u$, we need to satisfy the constraint $\langle \mu, \mathbf{1} \rangle = u$.
- If $\langle \nu, \mathbf{1} \rangle < l$, we need to satisfy the constraint $\langle \mu, \mathbf{1} \rangle = l$.

The last two cases correspond to solving the problem

$$\underset{\mu \in \mathbb{R}^k}{\operatorname{argmin}} \langle \mu, \log \mu \rangle - \langle \mu, \theta \rangle \quad \text{s.t.} \quad \langle \mu, \mathbf{1} \rangle = m, \quad \mu \le 1.$$

with $m = u$ or $m = l$. We can rewrite it as

$$\underset{\alpha \in \mathbb{R}^k}{\operatorname{argmin}} \langle \alpha, \log \alpha \rangle - \langle \alpha, z \rangle \quad \text{s.t.} \quad \langle \alpha, \mathbf{1} \rangle = 1, \quad \alpha \le \frac{1}{m}.$$

with $\alpha := \frac{\mu}{m}$ and $z := \theta - (\log m)\mathbf{1}$. An $O(k \log k)$ algorithm for solving this constrained softmax (KL projection onto a capped simplex) was derived in [33]. A related projection using a different entropy is derived in [3].

## C.3 Vertices of the order simplex (Proposition 4)

Let us gather the vertices $\varphi(y) \in \{0,1\}^{k-1}$ for all $y \in [k]$ as columns in a matrix $M \in \{0,1\}^{k-1 \times k}$. For instance, with $k = 4$,

$$M = \begin{bmatrix} 0 & 1 & 1 & 1 \\ 0 & 0 & 1 & 1 \\ 0 & 0 & 0 & 1 \end{bmatrix}.$$

Recall that

$$\mathcal{M} = M \triangle^k = \{Mp \colon p \in \mathbb{R}^k, p \ge 0, \langle p, \mathbf{1} \rangle = 1\} \subset \mathbb{R}^{k-1}.$$

Let $\mu = Mp$. Then for all $p \in \triangle^k$

$$\mu_1 = p_2 + p_3 + \cdots + p_k$$
$$\mu_2 = p_3 + p_4 + \cdots + p_k$$
$$\mu_3 = p_4 + \cdots + p_k$$
$$\vdots$$
$$\mu_{k-1} = p_k,$$

from which we obtain

$$1 - \mu_1 = p_1 \ge 0$$
$$\mu_1 - \mu_2 = p_2 \ge 0$$
$$\mu_2 - \mu_3 = p_3 \ge 0$$
$$\vdots$$
$$\mu_{k-1} = p_k \ge 0.$$

Notice that $\langle p, \mathbf{1} \rangle = 1$ is automatically satisfied for any $\mu = Mp$. Therefore

$$\mathcal{M} = \{\mu \in \mathbb{R}^{k-1} \colon 1 \ge \mu_1 \ge \mu_2 \ge \cdots \ge \mu_{k-1} \ge 0\},$$

which is known as the order simplex [22].

## C.4 Calibration of target and surrogate excess risks (Proposition 5)

We prove Proposition 5, extending a recent analysis [38] to the more general projection losses.

### C.4.1 Background

In this section, after reviewing the classical notions of pointwise and population excess risks, we discuss calibration functions for structured prediction, as introduced in [40]. We use the generalized notation introduced in §A, with output space $\mathcal{O}$ and ground truth space $\mathcal{Y}$.

**Pointwise and population risks.** Given a distribution $q \in \triangle^{|\mathcal{Y}|}$, we define the *pointwise* risk of $\widehat{y} \in \mathcal{O}$ for the loss $L$ and the pointwise risk of $\theta \in \Theta$ for the surrogate $S$ by

$$\ell(\widehat{y}, q) := \mathbb{E}_{Y \sim q}\, L(\widehat{y}, Y) \quad \text{and} \quad s(\theta, q) := \mathbb{E}_{Y \sim q}\, S(\theta, Y),$$

respectively. We also define the corresponding excess of pointwise risks, the difference between the pointwise risks and the pointwise Bayes risk:

$$\delta\ell(\widehat{y}, q) := \ell(\widehat{y}, q) - \inf_{y' \in \mathcal{O}} \ell(y', q) \quad \text{and} \quad \delta s(\theta, q) := s(\theta, q) - \inf_{\theta' \in \Theta} s(\theta', q).$$

Given a joint distribution $p \in \triangle(\mathcal{X} \times \mathcal{Y})$, let us now define the *population* target risk and the population surrogate risk by

$$\mathcal{L}(f) := \mathbb{E}_{(X,Y) \sim p}\, L(f(X), Y) \quad \text{and} \quad \mathcal{S}(g) := \mathbb{E}_{(X,Y) \sim p}\, S(g(X), Y).$$

The quality of estimators $f$ and $g$ is measured in terms of the excess of population risks

$$\delta\mathcal{L}(f) := \mathcal{L}(f) - \inf_{f' : \mathcal{X} \to \mathcal{O}} \mathcal{L}(f') \quad \text{and} \quad \delta\mathcal{S}(g) := \mathcal{S}(g) - \inf_{g' : \mathcal{X} \to \Theta} \mathcal{S}(g').$$

Note that the population risks can be written in terms of the pointwise ones as

$$\mathcal{L}(f) = \mathbb{E}_{X \sim p_{\mathcal{X}}}\, \ell(f(X), p(\cdot|X)) \quad \text{and} \quad \mathcal{S}(g) = \mathbb{E}_{X \sim p_{\mathcal{X}}}\, s(g(X), p(\cdot|X)), \tag{15}$$

where $p(\cdot|x)$ is the conditional distribution over $\mathcal{Y}$, and $p_{\mathcal{X}}$ is the marginal distribution over $\mathcal{X}$. Analogously, when the surrogate is a F-Y loss generated by $\Omega$, we will use $s_\Omega$, $\delta s_\Omega$, $\mathcal{S}_\Omega$ and $\delta\mathcal{S}_\Omega$.

**Calibration functions.** Let $d \colon \Theta \to \mathcal{O}$ be a decoding function, namely a function that turns a continuous prediction $\theta = g(x)$ into a discrete structure in $\mathcal{O}$. A calibration function $\zeta$ is a function relating the excess of *pointwise* risks $\delta\ell$ and $\delta s$ for all $\theta \in \Theta$ and $q \in \triangle^{|\mathcal{Y}|}$ by

$$\zeta(\delta\ell(d(\theta), q)) \leq \delta s(\theta, q).$$

It allows to control how much reduction of $\delta s$ is needed to reduce $\delta\ell$ when using $d$ as a decoder (larger $\zeta$ is better). As shown in [40], $\zeta$ can be cast as an optimization problem,

$$\zeta(\epsilon) = \inf_{\theta \in \Theta, q \in \triangle^{|\mathcal{Y}|}} \delta s(\theta, q) \quad \text{s.t.} \quad \delta\ell(d(\theta), q) \geq \epsilon. \tag{16}$$

It is easy to verify that $\zeta$ is positive, non-decreasing, and satisfies $\zeta(0) = 0$. As shown in [40, 38], any convex lower-bound $\xi$ of $\zeta$ allows to in turn calibrate the excess of *population* risks:

$$\xi(\delta\mathcal{L}(d \circ g)) \leq \delta\mathcal{S}(g), \tag{17}$$

for all $g \colon \mathcal{X} \to \Theta$ and $d \colon \Theta \to \mathcal{O}$. This follows from Jensen's inequality and from (15). If $\xi(\epsilon) > 0$ for all $\epsilon > 0$ and $\xi(0) = 0$, this implies Fisher consistency.

### C.4.2 Calibration function of Fenchel-Young losses

We derive the exact expression of the calibration function (16) for **general** Fenchel-Young losses.

---

**Lemma 1** *Calibration function of general Fenchel-Young losses*

*Let $L(\widehat{y}, y)$ be decomposed as (12) and $S(\theta, y) = S_\Omega(\theta, \varphi(y)) := \Omega^*(\theta) + \Omega(\varphi(y)) - \langle \theta, \varphi(y) \rangle$, with $\varphi(\mathcal{Y}) \subseteq \mathrm{dom}(\Omega)$. Then the calibration function (16) with decoder $d \colon \Theta \to \mathcal{O}$ reads*

$$\zeta(\epsilon) = \inf_{\theta \in \Theta, \mu \in \mathcal{M}} S_\Omega(\theta, \mu) \quad \text{s.t.} \quad \langle \psi(d(\theta)) - \psi(\widehat{y}_L(\mu)), V\mu + b \rangle \geq \epsilon.$$

---

**Proof.** Let $\mu_\varphi(q) := \mathbb{E}_{Y\sim q}[\varphi(Y)]$. The pointwise surrogate risk reads

$$
\begin{aligned}
s_\Omega(\theta, q) &:= \mathbb{E}_{Y\sim q}[S_\Omega(\theta, \varphi(Y))] \\
&= \sum_{y\in\mathcal{Y}} q(y)(\Omega^*(\theta) + \Omega(\varphi(y)) - \langle\theta, \varphi(y)\rangle) \\
&= \Omega^*(\theta) + \Omega(\mu_\varphi(q)) - \langle\theta, \mu_\varphi(q)\rangle + \mathbb{E}_{Y\sim q}[\Omega(\varphi(Y))] - \Omega(\mu_\varphi(q)) \\
&=: S_\Omega(\theta, \mu_\varphi(q)) + I_\Omega(\varphi(Y), q),
\end{aligned}
$$

where we defined $I_\Omega(\varphi(Y), q)$, the Bregman information [5] of the random variable $\varphi(Y)$ with generating function $\Omega$. Hence the excess of pointwise surrogate risk reads

$$
\begin{aligned}
\delta s_\Omega(\theta, q) &= s_\Omega(\theta, q) - \inf_{\theta'\in\Theta} s_\Omega(\theta', q) \\
&= S_\Omega(\theta, \mu_\varphi(q)) - \underbrace{\inf_{\theta'\in\Theta} S_\Omega(\theta', \mu_\varphi(q))}_{=0} \\
&= S_\Omega(\theta, \mu_\varphi(q)),
\end{aligned}
$$

where in the second line we used [11, Proposition 2]. The pointwise target risk reads

$$
\begin{aligned}
\ell(\widehat{y}, q) &= \mathbb{E}_{Y\sim q}[L(\widehat{y}, Y)] \\
&= \sum_{y\in\mathcal{Y}} q(y)((\langle\psi(\widehat{y}), V\varphi(y) + b\rangle + c(y)) \\
&= \langle\psi(\widehat{y}), V\mu_\varphi(q) + b\rangle + \mathbb{E}_{Y\sim q}[c(Y)].
\end{aligned}
$$

Hence the excess of pointwise target risk reads

$$
\begin{aligned}
\delta\ell(\widehat{y}, q) &= \ell(\widehat{y}, q) - \inf_{y'\in\mathcal{O}} \ell(y', q) \\
&= \langle\psi(\widehat{y}), V\mu_\varphi(q) + b\rangle - \inf_{y'\in\mathcal{O}} \langle\psi(y'), V\mu_\varphi(q) + b\rangle \\
&= \langle\psi(\widehat{y}), V\mu_\varphi(q) + b\rangle - \langle\psi(\widehat{y}_L(\mu_\varphi(q))), V\mu_\varphi(q) + b\rangle \\
&= \langle\psi(\widehat{y}) - \psi(\widehat{y}_L(\mu_\varphi(q))), V\mu_\varphi(q) + b\rangle,
\end{aligned}
$$

where

$$
\widehat{y}_L(u) := \operatorname*{argmin}_{y'\in\mathcal{O}} \langle\psi(y'), Vu + b\rangle.
$$

Therefore we can rewrite (16) as

$$
\zeta(\epsilon) = \inf_{\theta\in\Theta, q\in\triangle^{|\mathcal{Y}|}} S_\Omega(\theta, \mu_\varphi(q)) \quad \text{s.t.} \quad \langle\psi(d(\theta)) - \psi(\widehat{y}_L(\mu_\varphi(q))), V\mu_\varphi(q) + b\rangle \geq \epsilon.
$$

Using the change of variable $\mu = \mu_\varphi(q) \in \mathcal{M} = \operatorname{conv}(\varphi(\mathcal{Y}))$ gives the desired result. $\square$

### C.4.3 Technical lemma

We give in this section a technical lemma, which will be useful for the rest of the proof.

---

**Lemma 2** *Upper-bound on pointwise target risk*

*Let $\sigma := \sup_{\widehat{y}\in\mathcal{O}} \|V^\top\psi(\widehat{y})\|_*$, where $\|\cdot\|_*$ denotes the dual norm of $\|\cdot\|$. Then,*

$$
\delta\ell(\widehat{y}_L(u), q) \leq 2\sigma\|\mu_\varphi(q) - u\| \quad \forall u \in \mathbb{R}^p, q \in \triangle^{|\mathcal{Y}|}.
$$

---

**Proof.** The proof is a slight modification of [38, Lemma D.3] and is included for completeness. In that work, $V = I$ and $b = 0$, or put differently, they are absorbed into $\varphi$. In this work, we keep $V$ and $b$ explicitly to decouple the label encoding from the loss decomposition. This is important in order to keep MAP and projection algorithms unchanged. Following [38], let us decompose $\delta\ell(\widehat{y}_L(u), q)$ into two terms $A$ and $B$:

$$
\begin{aligned}
\delta\ell(\widehat{y}_L(u), q) &= \langle\psi(\widehat{y}_L(u)) - \psi(\widehat{y}_L(\mu_\varphi(q))), V\mu_\varphi(q) + b\rangle \\
&= \underbrace{\langle\psi(\widehat{y}_L(u)), V\mu_\varphi(q) + b - Vu - b\rangle}_{A} + \underbrace{\langle\psi(\widehat{y}_L(u)), Vu + b\rangle - \langle\psi(\widehat{y}_L(\mu_\varphi(q))), V\mu_\varphi(q) + b\rangle}_{B}.
\end{aligned}
$$

Clearly, $A \leq \sup_{\widehat{y} \in \mathcal{O}} |\langle \psi(\widehat{y}), V\mu_\varphi(q) - Vu \rangle|$.

Using $|\min_z \eta(z) - \min_z \eta'(z)| \leq \sup_z |\eta(z) - \eta'(z)|$, an inequality also used in [15, Thm. 12], we also get $B \leq \sup_{\widehat{y} \in \mathcal{O}} |\langle \psi(\widehat{y}), V\mu_\varphi(q) - Vu \rangle|$.

Therefore, in both cases, we see that $b$ cancels out. Combining the two, we obtain
$$\delta\ell(\widehat{y}_L(u), q) \leq 2 \sup_{\widehat{y} \in \mathcal{O}} |\langle \psi(\widehat{y}), V\mu_\varphi(q) - Vu \rangle| = 2 \sup_{\widehat{y} \in \mathcal{O}} |\langle V^\top \psi(\widehat{y}), \mu_\varphi(q) - u \rangle|.$$

By definition of the dual norm, we have the (generalized) Cauchy-Shwarz inequality $\langle x, y \rangle \leq \|x\| \, \|y\|_*$. Therefore,
$$\delta\ell(\widehat{y}_L(u), q) \leq 2 \sup_{\widehat{y} \in \mathcal{O}} |\langle V^\top \psi(\widehat{y}), \mu_\varphi(q) - u \rangle| \leq 2 \sup_{\widehat{y} \in \mathcal{O}} \|V^\top \psi(\widehat{y})\|_* \|\mu_\varphi(q) - u\|.$$

$\square$

### C.4.4 Convex lower bound on the calibration function of projection-based losses

Since $\zeta$ above could be non-convex and difficult to compute, we next derive a convex lower bound for a particular **subset** of Fenchel-Young losses (namely, projection-based losses $S_\mathcal{C}^\Psi$) and for a particular decoder $d \colon \Theta \to \mathcal{O}$ (namely, $d = \widehat{y}_L \circ P_\mathcal{C}^\Psi$).

---

**Lemma 3** *Convex lower bound on the calibration function of $S_\mathcal{C}^\Psi$*

*Let $L(\widehat{y}, y)$ be decomposed as (12) and $S(\theta, y) = S_\mathcal{C}^\Psi(\theta, y)$ be defined as in (6). Assume $\Psi$ is $\frac{1}{\beta}$-strongly convex over $\mathcal{C}$ w.r.t. $\|\cdot\|$, Legendre-type, and $\mathcal{C}$ is a convex set such that $\varphi(\mathcal{Y}) \subseteq \mathcal{C} \subseteq \mathrm{dom}(\Psi)$. Let $\sigma := \sup_{\widehat{y} \in \mathcal{O}} \|V^\top \psi(\widehat{y})\|_*$, where $\|\cdot\|_*$ is the dual norm of $\|\cdot\|$. Then, the calibration function defined (11) with $d = \widehat{y}_L \circ P_\mathcal{C}^\Psi$ is lower bounded as*

$$\zeta(\epsilon) \geq \frac{\epsilon^2}{8\beta\sigma^2}.$$

---

**Proof.** Let us set $\Omega = \Psi + I_\mathcal{C}$, where $\Psi$ is Legendre-type. Note that this does not imply that $\Omega$ itself is Legendre-type. Using Lemma 2, we have for all $u \in \mathbb{R}^p$ and all $q \in \triangle^{|\mathcal{Y}|}$
$$\delta\ell(\widehat{y}_L(u), q) \leq 2\sigma\|\mu_\varphi(q) - u\|.$$

Let us set the decoder to $d = \widehat{y}_L \circ \nabla\Omega^*$. With $u = \nabla\Omega^*(\theta)$, we thus get for all $\theta \in \Theta$ and $q \in \triangle^{|\mathcal{Y}|}$:
$$\epsilon \leq \delta\ell(d(\theta), q) \leq 2\sigma\|\mu_\varphi(q) - \nabla\Omega^*(\theta)\|.$$

From (13), $\Psi$ is $\frac{1}{\beta}$-strongly convex over $\mathcal{C}$ w.r.t. $\|\cdot\|$ if and only if for all $u, v \in \mathcal{C}$
$$D_\Psi(u, v) \geq \frac{1}{2\beta}\|u - v\|^2.$$

Combining this with [11, Proposition 3], we have for all $\theta \in \Theta$ and all $u \in \mathcal{C}$
$$S_\Omega(\theta, u) \geq D_\Psi(u, \nabla\Omega^*(\theta)) \geq \frac{1}{2\beta}\|u - \nabla\Omega^*(\theta)\|^2.$$

Altogether, we thus get for all $\theta \in \Theta$ and $q \in \triangle^{|\mathcal{Y}|}$
$$\begin{aligned}
\delta s_\Omega(\theta, q) &= S_\Omega(\theta, \mu_\varphi(q)) \\
&\geq D_\Psi(\mu_\varphi(q), \nabla\Omega^*(\theta)) \\
&\geq \frac{1}{2\beta}\|\mu_\varphi(q) - \nabla\Omega^*(\theta)\|^2 \\
&\geq \frac{1}{8\beta\sigma^2}\delta\ell(d(\theta), q)^2 \\
&\geq \frac{\epsilon^2}{8\beta\sigma^2}.
\end{aligned}$$

$\square$

### C.5 Finalizing the proof

We simply combine (17) and Lemma 3. $\square$