[Reviews · NeurIPS 2019]

Reviewer 1



Post-feedback update: Thanks for your update. Your additional explanations and results will help improve the paper, and I definitely think this work is strong and should be accepted. -------------------------------------------------------------------------------------------------------- Originality: This paper draw upon a lot of previous work studying losses within structured prediction, all of which is cited throughout in appropriate places. The framework itself is new, and the authors make it very clear how prior work fits into the framework as special cases. At the same time, a good case is made for why this framework is useful to have and how it can be better to use than prior losses. Quality: this paper makes a compelling case for the framework it introduces. It explains how previously developed structured prediction losses fit into it as well as providing examples for additional losses that can be derived and how they might be better to due to their ability to be computed more cheaply. A consistency bound is proven, indicating that using this framework is a sensible thing to do. A tradeoff between computational cost and statistical efficiency is mentioned at several points throughout the paper; however, the experimental results do not reflect on these tradeoffs, as runtime comparisons for both training and inference are not included. The experiments do adequately demonstrate, though, that the use of some of the losses introduced by this framework that are not subsumed by older frameworks can lead to improved models. Clarity: Overall, the flow of ideas in this paper is quite clear. Background material is described in sufficient detail for the reader to understand necessary concepts and how they relate to what is presented here. Section 3.2 is very helpful for understanding examples of the sorts of losses that can be constructed using this framework. There are a few minor typos (for example, in line 101, the first Psi function definition is missing its argument), but these do not significantly impede the clarity of the work. Significance: the provided empirical results demonstrate that practitioners of structured prediction should consider thinking about developing losses using this framework, as they may lead to training models with better performance. Because no information is provided on the relative runtimes of the compared approaches, the tradeoffs between model performance and training/inference time are not as clear as they could be.

Reviewer 2



Overview: ------------- This paper proposes and studies a general framework for generating surrogate loss functions for structured prediction problems. In particular, the loss is defined in terms of Bregman projections, and some examples are given for when the projection (or a relaxation thereof) can be computed efficiently. Surrogates in this family are shown to be calibrated wrt the true loss. This paper generalizes SparseMAP [26] and Fenchel-Young losses [6] to Bregman projections. It seems to make an interesting contribution to the field of structured prediction, by proposing a family of calibrated surrogate losses. Limitations of the paper include questionable practicality of the approach (due to expensive training and inference), and proximity to previous work [6,26] which is generalized (the calibration results here seem novel). Comments/questions: ----------------------------- Since computational-statistical trade-off is central to the paper, consider reporting runtimes in experiments so the reader can get a sense of what this looks like for particular problem instances. Further, I think a simple structured-prediction baseline (pick your favorite, maybe SPENs?) is missing in the experiments in order to get a sense of how the proposed framework compares to existing approaches in terms of accuracy and complexity (the only comparison is between variants of the proposed method). At the end of section 4 results are described for projected averaged SGD, but experimental results are reported for L-BFGS. What is the reason for this discrepancy? For multilable classification, can you comment on calibration of the losses used here w.r.t. the F1 measure? Might be good to explicitly include somewhere the general training algorithm (minimize the surrogate S). Haven’t seen it anywhere. Minors: ---------- - Line 19: not sure I understand the statement saying structured objects are more difficult than mere vectors. It seems like the structure should actually help learning since otherwise every vector output is equally possible. - Line 78: not tractable => intractable - Line 101: “\Psi = 1/2||u||” => “\Psi(u) = 1/2||u||” - Can you please explain the inequalities in line 114? Psi and Omega are functions, so does it mean that this holds for any input? - Line 133: “we will we” - Line 198: would be good to mention a couple of common losses here in addition to referring to the appendix. - Lines 268, 275: better mention that Tables 3, 5, 6 are in appendix B.

Reviewer 3



Originality: The idea is not entirely new. L106 gives an interesting explanation of F-Y loss, which seems to be a novel perspective. Basically, the paper focuses on a special case of the F-Y loss [6] for structured prediction; the consistency analysis of excess Bayes surrogate risk is based on the work of [27, 29] using calibration function. Quality: All proofs look good to me (after a quick check). The experimental part of this paper is relatively weak: it only shows the proposed F-Y losses work for selective examples. However, there is no comparison with prior works. Classical baselines such as SSVM, CRF, SparseMAP etc should be compared. Clarity: The paper is generally well written with a clear structure. Significance: Convex surrogates such as F-Y loss should be interesting to structured prediction community. This paper may attract more people (potentially deep learning folks) exploring better surrogates/pipelines for structured prediction.

[Author Response · NeurIPS 2019]

We are grateful to our three reviewers for their time and insightful comments.

## Reviewer 2

**"a runtime comparison is needed to understand tradeoffs btw the different choices of projection/decoding sets"**

This is a great remark, that we will adress. For decoding, we believe it is important to use the marginal polytope (convex hull) whenever possible, to make sure to predict valid structures. For projections, the knapsack polytope, order simplex and permutahedron have (near) linear-time algorithms so the projection step is very fast. The only potentially problematic case is the Birkhoff polytope, whose projection takes quadratic time. However, we found that the runtimes were surprisingly fast in our experiments. Moreover, quadratic time is still better than the CRF loss, which is intractable. Our provided implementation backs up our claim. We will also report precise runtime figures in the final version.

**"Having experiments involving more types of models than just linear [...] could further improve significance"**

We focused on linear models to remain in the realm of convex optimization but exploring more models (e.g., neural networks, gradient boosting) would be very interesting indeed. Note however that $\frac{1}{2}\|\varphi(y) - \nabla\Omega^*(\theta)\|^2 \leq \frac{1}{2}\|\varphi(y) - \theta\|^2$ for all $\theta \in \mathbb{R}^d$ and $y \in \mathcal{Y}$ so the Euclidean projection $\nabla\Omega^*(\theta)$ theoretically achieves smaller loss than $\theta = g(x)$ alone, regardless of the model. This is also true for the more general Bregman projection setting.

## Reviewer 3

**"Limitations include questionable practicality of the approach (due to expensive training and inference) and proximity to previous work [6,26] which is generalized"**

Inference / decoding is unavoidable and is used in virtually all structured prediction approaches. The projection step during training takes linear time for the knapsack polytope, order simplex and permutahedron. The only potentially problematic case is the Birkhoff polytope, whose projection takes quadratic time but this is still better than the CRF loss, which is intractable.

Besides consistency analysis, our contributions include the idea that we can use any convex set (and not just the marginal polytope), Prop. 2 which guarantees smoothness of our loss in the KL projection case and Prop. 3 on the order simplex.

**"What is the reason for this discrepancy?" (ASGD vs. LBFGS)**

ASGD is convenient to analyze when the objective is an expectation (as is the case of the surrogate risk) but in practice we can use any ERM algorithm. We chose LBFGS as it does not require choosing a learning rate.

**"Can you comment on calibration of the losses used here w.r.t. the F1 measure?"**

For multilabel classification, the unit cube and knapsack polytope lead to losses that are consistent with accuracy (Hamming loss) but there is currently no guarantee for $F_1$ score. Deriving a polytope and a projection which guarantees calibration w.r.t. $F_1$ score is actually a great open question we would like to tackle in the future.

Thank you very much for all your other comments, we will address them.

## Reviewer 4

**"Classical baselines such as SSVM, CRF, SparseMAP etc should be compared."**

In our experiments, we specifically chose tasks for which the CRF loss is not available. For instance, for permutation problems, the CRF loss is intractable, as marginal inference is #P-complete. For multilabel prediction with cardinality constraints, there does not exist any prior algorithm for computing the CRF loss and tractability is an open question.

SparseMAP is already included in our experiments as it corresponds to using Euclidean projections on the marginal polytope (Birkhoff polytope for ranking, knapsack polytope for classification, order simplex for ordinal regression).

A comparison with SSVM would indeed be possible and interesting, although the loss is not consistent and not smooth.

**"L100–L106 are key to the paper but are not clearly explained. The technical details should be elaborated [...]**

We agree that we were too short on this. We plan to add more details in the final version using the ninth page.

**"the influence of different surrogates to the performance is less important than that of the choice of decoding space? E.g., comparing Simplex+B with B+B. Could you make a comment on this?"**

It is true that the decoding over $\mathcal{B}$ goes a long way, since it makes sure that a valid permutation matrix is always predicted. But we argue that Proposition 1 is insightful: even using $[0,1]^k$ instead $\mathbb{R}^k$ improves test accuracy a lot, according to our experiments. Regarding $\mathcal{B}$ vs. $\triangle$, we argue that our empirical results are convincingly in favor of $\mathcal{B}$. Also, when using $\triangle$, we lose the ability to predict soft permutations as shown in Figure 2.

[Meta-Review · NeurIPS 2019]

All reviewers agreed that this paper make a nice contribution to NeurIPS by providing a novel general framework for generating calibrated surrogate loss functions for structured prediction problems. On the other hand, in discussion, they also stressed that including some baselines (e.g., SSVM/CRF+approximation/SPEN) in the experiments and reporting runtimes could make this paper much stronger. The authors should implement their promised changes in the camera-ready version.